# Deep Learning Models for Poorly Differentiated Colorectal Adenocarcinoma Classification in Whole Slide Images Using Transfer Learning

**DOI:** 10.3390/diagnostics11112074

**Published:** 2021-11-09

**Authors:** Masayuki Tsuneki, Fahdi Kanavati

**Affiliations:** Medmain Research, Medmain Inc., Fukuoka 810-0042, Japan; fkanavati@medmain.com

**Keywords:** deep learning, transfer learning, poorly differentiated adenocarcinoma, colon

## Abstract

Colorectal poorly differentiated adenocarcinoma (ADC) is known to have a poor prognosis as compared with well to moderately differentiated ADC. The frequency of poorly differentiated ADC is relatively low (usually less than 5% among colorectal carcinomas). Histopathological diagnosis based on endoscopic biopsy specimens is currently the most cost effective method to perform as part of colonoscopic screening in average risk patients, and it is an area that could benefit from AI-based tools to aid pathologists in their clinical workflows. In this study, we trained deep learning models to classify poorly differentiated colorectal ADC from Whole Slide Images (WSIs) using a simple transfer learning method. We evaluated the models on a combination of test sets obtained from five distinct sources, achieving receiver operating characteristic curve (ROC) area under the curves (AUCs) up to 0.95 on 1799 test cases.

## 1. Introduction

According to global cancer statistics in 2020 [1,2], colorectal cancer (CRC) is amongst the most common leading causes of cancer deaths in the world and the second most common in the United States [1]. In 2020, approximately 147,950 individuals were diagnosed with CRC, and 53,200 died from the disease. Screening and high-quality treatment can help in improving survival prospects. This is evidenced by the decrease in CRC death rates from 2008 to 2017 by 3% annually in individuals aged 65 years and older and by 0.6% annually in individuals aged 50 to 64 years. However, as the incidence of CRC in young and middle-aged adults (younger than 50 years) has been increasing (by 1.3% annually [1]), the American Cancer Society lowered the recommended age for screening initiation for individuals at average risk from 50 to 45 years in 2018 [2,3]. The development of endoscopy provided a major impact on the diagnosis and treatment of CRC, especially colonoscopy which allows observation of the colonic mucosal surface with biopsies of identified and/or suspicious lesions [4].

Deep learning has been successfully applied in computational pathology in the past few years for tasks such cancer classification, cell segmentation, and outcome prediction for a variety of organs and diseases [5,6,7,8,9,10,11,12,13,14,15,16,17,18]. For the classification of tumour in colorectal WSI, Iizuka et al. [18] trained a model on a large dataset of colorectal WSIs for the classification of well differentiated ADC; however, it did not include poorly differentiated ADC.

Histopathologically, adenocarcinoma (ADC) account for more than 90% of CRCs. Most colorectal ADCs are well to moderately differentiated types with gland-forming and configuration of the glandular structures. If more than 50% of the tumor is formed by non-gland forming carcinoma cells, the tumor is classified as poorly differentiated (or high grade) ADC [19,20,21]. The frequency of poorly differentiated ADC is relatively low (3.3% to 18% of all CRCs) [22,23,24,25,26]. The 5-year survival rate for patients with colorectal poorly differentiated ADC is 20% to 45.5%, which indicates that poorly differentiated ADC has a less favorable prognosis compared with that of well or moderately differentiated ADC [23,26,27,28].

In this paper, we trained deep learning models for the classification of diffuse-type ADC in endoscopic biopsy specimen whole slide images (WSIs). We have used the partial transfer learning method [29] to fine-tune the models. We obtained models with ROC AUCs up to 0.95 for on the combined test set with a total of 1799 WSIs, demonstrating the potential of such methods for aiding pathologists in their workflows.

## 2. Methods

### 2.1. Clinical Cases and Pathological Records

For the present retrospective study, a total of 2547 endoscopic biopsy cases of human colorectal epithelial lesions HE (hematoxylin & eosin) stained histopathological specimens were collected from the surgical pathology files of five hospitals: International University of Health and Welfare, Mita Hospital (Tokyo) and Kamachi Group Hospitals (consisting of Wajiro, Shinmizumaki, Shinkomonji, and Shinyukuhashi hospitals) (Fukuoka) after histopathological review of those specimens by surgical pathologists. The experimental protocol was approved by the ethical board of the International University of Health and Welfare (No. 19-Im-007) and Kamachi Group Hospitals. All research activities complied with all relevant ethical regulations and were performed in accordance with relevant guidelines and regulations in the all hospitals mentioned above. Informed consent to use histopathological samples and pathological diagnostic reports for research purposes had previously been obtained from all patients prior to the surgical procedures at all hospitals, and the opportunity for refusal to participate in research had been guaranteed by an opt-out manner. The test cases were selected randomly, so the obtained ratios reflected a real clinical scenario as much as possible. All WSIs were scanned at a magnification of ×20, and the average dimension was 30 K × 15 K pixels. This protocol is similar to [18,30,31,32].

### 2.2. Dataset and Annotations

Prior to this study, the diagnosis of each WSI was verified by at least two pathologists, and they excluded cases that were inappropriate or of poor scanned quality. In particular, about 20% of poorly differentiated ADC cases were excluded due to disagreement between pathologists. Table 1 breaks down the distribution of the dataset into training, validation, and test sets. Hospitals which provided histopathological cases were anonymised (e.g., Hospital 1–5). The training and test sets were solely composed of WSIs of endoscopic biopsy specimens. The patients’ pathological records were used to extract the WSIs’ pathological diagnoses. In total, 36 WSIs from the training and validation sets had a poorly differentiated ADC diagnosis. They were manually annotated by a group of two surgical pathologists who perform routine histopathological diagnoses. The pathologists carried out detailed cellular-level annotations by free-hand drawing around poorly differentiated ADC cells. The well to moderately differentiated ADC (*n* = 71), adenoma (*n* = 110) and non-neoplastic subsets (*n* = 531) of the training and validation sets were not annotated and the entire tissue areas within the WSIs were used. Each annotated WSI was observed by at least two pathologists, with the final checking and verification performed by a senior pathologist. This dataset preparation is similar to [18,30,31,32].

### 2.3. Deep Learning Models

We trained all the models using the partial fine-tuning approach [29]. This method simply consists of using the weights of an existing pre-trained model and only fine-tuning the affine parameters of the batch normalisation layers and the final classification layer. We used the EfficientNetB1 [33] model starting with pre-trained weights on ImageNet. The total number of trainable parameters was only 63,329.

The training method that we have used in this study is exactly the same as reported in a previous study [30]. For completeness, we repeat the method here. To apply the CNN on the WSIs, we performed slide tiling by extracting square tiles from tissue regions. On a given WSI, we detected the tissue regions and eliminated most of the white background by performing a thresholding on a grayscale version of the WSI using Otsu’s method [34]. During prediction, we performed the tiling in a sliding window fashion, using a fixed-size stride, to obtain predictions for all the tissue regions. During training, we initially performed random balanced sampling of tiles from the tissue regions, where we tried to maintain an equal balance of each label in the training batch. To do so, we placed the WSIs in a shuffled queue such that we looped over the labels in succession (i.e., we alternated between picking a WSI with a positive label and a negative label). Once a WSI was selected, we randomly sampled $\frac{\mathrm{batch}\;\mathrm{size}}{\mathrm{num}\;\mathrm{labels}}$ tiles from each WSI to form a balanced batch. To maintain the balance on the WSI, we oversampled from the WSIs to ensure the model trained on tiles from all of the WSIs in each epoch. We then switched into hard mining of tiles once there was no longer any improvement on the validation set after two epochs. To perform the hard mining, we alternated between training and inference. During inference, the CNN was applied in a sliding window fashion on all of the tissue regions in the WSI, and we then selected the *k* tiles with the highest probability for being positive if the WSI was negative and the *k* tiles with the lowest probability for being positive if the WSI was positive. This step effectively selected the hard examples with which the model was struggling. The selected tiles were placed in a training subset, and once that subset contained *N* tiles, the training was run. We used k=16, N=256, and a batch size of 32.

From the WSIs with poorly-differentiated ADC, we sampled tiles based on the free-hand annotations. If the WSI contained annotations for cancer cells, then we only sampled tiles from the annotated regions as follows: if the annotation was smaller than the tile size, then we sampled the tile at the centre of the annotation regions; otherwise, if the annotation was larger than the tile size, then we subdivided the annotated regions into overlapping grids and sampled tiles. Most of the annotations were smaller than the tile size. On the other hand, if the WSI did not contain diffuse-type ADC, then we freely sampled from the entire tissue regions.

The models were trained on WSIs at ×20 magnification. To obtain a prediction on a WSI, the model was applied in a sliding window fashion using a tile size of 512 × 512 px and a stride of 256 × 256 px, generating a prediction per tile. The WSI prediction was then obtained by taking the maximum from all of the tiles. The prediction output for the ensemble model was obtained as simply the average output of the three models used.

We trained the models with the Adam optimisation algorithm [35] with the following parameters: $beta_{1}=0.9$, $beta_{2}=0.999$, and a batch size of 32. We used a learning rate of 0.001 when fine-tuning. We applied a learning rate decay of 0.95 every 2 epochs. We used the binary cross entropy loss function. We used early stopping by tracking the performance of the model on a validation set, and training was stopped automatically when there was no further improvement on the validation loss for 10 epochs. The model with the lowest validation loss was chosen as the final model.

### 2.4. Software and Statistical Analysis

We implemented the models using TensorFlow [36]. We calculated the AUCs in python using the scikit-learn package [37] and performed the plotting using matplotlib [38]. We performed image processing, such as the thresholding with scikit-image [39]. We computed the 95% CIs estimates using the bootstrap method [40] with 1000 iterations. We used openslide [41] to perform realtime slide tiling.

## 3. Results

The aim of this study was to train a deep learning model based on convolutional neural networks (CNNs) to classify poorly differentiated ADC in WSIs of colorectal biopsy specimens. We had a total of 748 WSIs which were available for training of which only 36 WSIs had poorly differentiated ADC. Given the small number of WSIs, we opted for using a transfer learning method which was suitable for such a task. Transfer learning consists of fine-tuning the weights of a model that was pre-trained on another dataset for which a larger number of images were available for training. To this end, we evaluated four models: (1) a model that was fine-tuned starting with pre-trained weights on ImageNet [42], (2) a model that was fine-tuned starting with pre-trained weights on a stomach WSIs dataset [30], (3) a model that was pre-trained for the classification of gastric poorly differentiated ADC which we did not fine-tune [30], and (4) a model which consisted of an ensemble [43] of the previous three models. Figure 1 shows an overview of our training method.

### Evaluation on Five Independent Test Sets from Different Sources

We evaluated our models on five distinct test sets consisting of biopsy specimens, three of which were from hospitals not in the training set. Table 1 breaks down the distribution of the WSIs in each test set. For each test set for for their combination, we computed the ROC AUC and log loss for the WSI classification of poorly differentiated ADC as well as the log loss, and we have summarised the results in Table 2 and Figure 2. Figure 3 and Figure 4 show true positive and false positive example heatmap outputs, respectively. Table 3 shows a confusion matrix for the combined test set using the ensemble model and a probability threshold of 0.5. When operating with a high sensitivity threshold (sensitivity 100%), the specificity for the best model on the combined test set was 75%.

## 4. Discussion

In this study, we trained deep learning models for the classification of poorly differentiated ADC from colorectal biopsy WSIs. We used transfer learning with a hard mining of false positives to train the models on a training set obtained from two hospitals. We evaluated four models—one of which was an ensemble of the other three—on five different test sets originating from different hospitals, and on the combination of all five, given the small number of cases with poorly differentiated ADC. Overall, we obtained high ROC AUC of about 0.95.

Given the histopathological similarity between gastric and colonic ADC, the stomach model—which was previously trained on poorly differentiated cases of gastric ADC—was still able to perform well on the colonic cases. There was improvement in further fine-tuning the model, where the ROC AUC increased from 0.89 to 0.95. However, the result is similar to having fine-tuned a model that was previously only trained on ImageNet. The ensemble model had similar AUC, albeit with the lowest log loss of all the models.

The primary limitation of this study was the small number of poorly differentiated ADC cases; overall, there were only 36 in the training set and 74 in the test set. The small number of poorly differentiated ADC is to be expected given that well to moderately differentiated ADC is typically more common. The combined test set contained a large number of well to moderately differentiated ADC and non-neoplastic, which increases the chances of false positives. Nonetheless, the models performed well and did not have a high false positive rate across a wide range of thresholds, based on the ROC curves in Figure 2.

Most ADCs in colon are moderately to well differentiated types. On the other hand, because poorly differentiated ADC exhibits the worst prognoses among the various types of colorectal cancer, it is important to classify on endoscopic biopsy specimens [26]. In this study, we have shown that deep learning models could potentially be used for the classification of poorly differentiated ADC. Using a simple transfer learning method, it was possible to train a high performing model relatively quickly compared to having to train a model from scratch. Deep learning models show high potential for aiding pathologists and improving the efficiency of their workflow systems.

Poorly differentiated ADC tends to grow and spread more quickly than well and moderately differentiated ADC, and this makes early screening critical for improving patient prognosis. The promising results of this study add to the growing evidence that deep learning models could be used as a tool to aid pathologists in their routine diagnostic workflows, potentially acting as a second screener. One advantage of using an automated tool is that it can systematically handle large amounts of WSIs without potential bias due to fatigue commonly experienced by surgical pathologists. It could also drastically alleviate the heavy clinical burden of daily pathology diagnosis. AI is considered a valuable tool that could transform the future of healthcare and precision oncology.

## Figures and Tables

**Figure 1 diagnostics-11-02074-f001:**
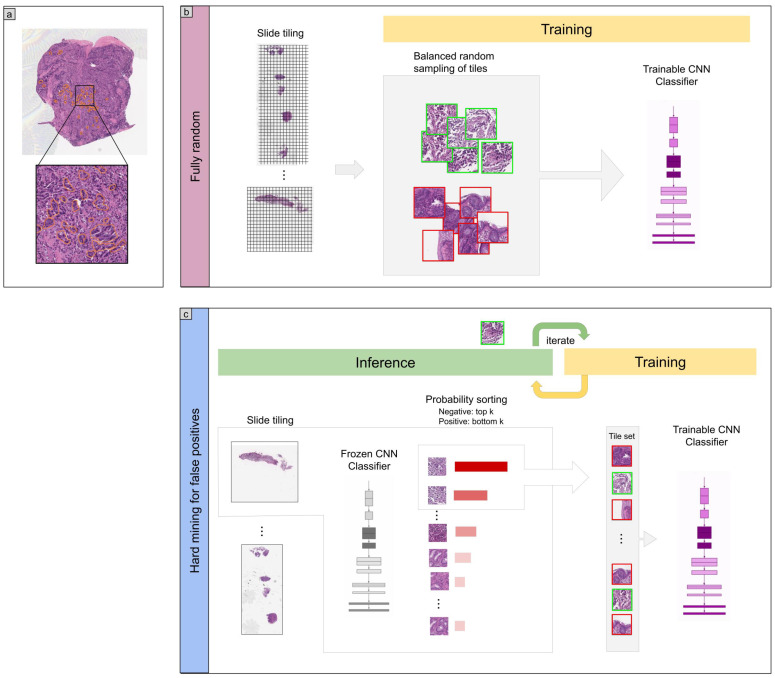
Method overview. (**a**) An example of poorly differentiated ADC annotation that was carried out on the WSIs by pathologists using in an in-house web-based application. (**b**) The initial training consisted in fully-random balanced sampling of positive (poorly differentiated ADC) and negative tiles to fine-tune the models. (**c**) After a few epochs of random sampling, the training switched into iterative hard mining of tiles that alternates between training and inference. During the inference step, we applied the model in a sliding window fashion on all of the WSI and selected the k tiles with the highest probabilities if the WSI was negative, and k tiles with the lowest probabilities if the WSI was positive. The tiles were collected in a subset that was batched and used for training. This process allows training to reduce false positives.

**Figure 2 diagnostics-11-02074-f002:**
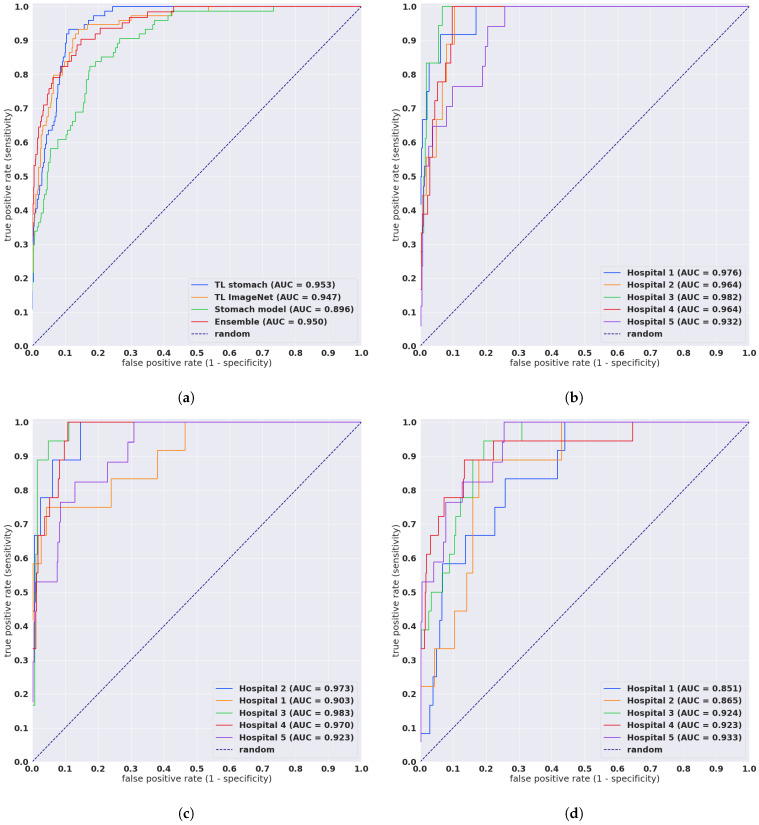
ROC curves for the four models as evaluated on the combined test sets and each test set separately. (**a**) Evaluation of the four models on the combined test set (*n* = 1741), (**b**) TL stomach model on the test sets that had poorly differentiated ADC, (**c**) TL ImageNet model on the test sets that had poorly ADC, (**d**) Stomach model on the test sets that had poorly differentiated ADC.

**Figure 3 diagnostics-11-02074-f003:**
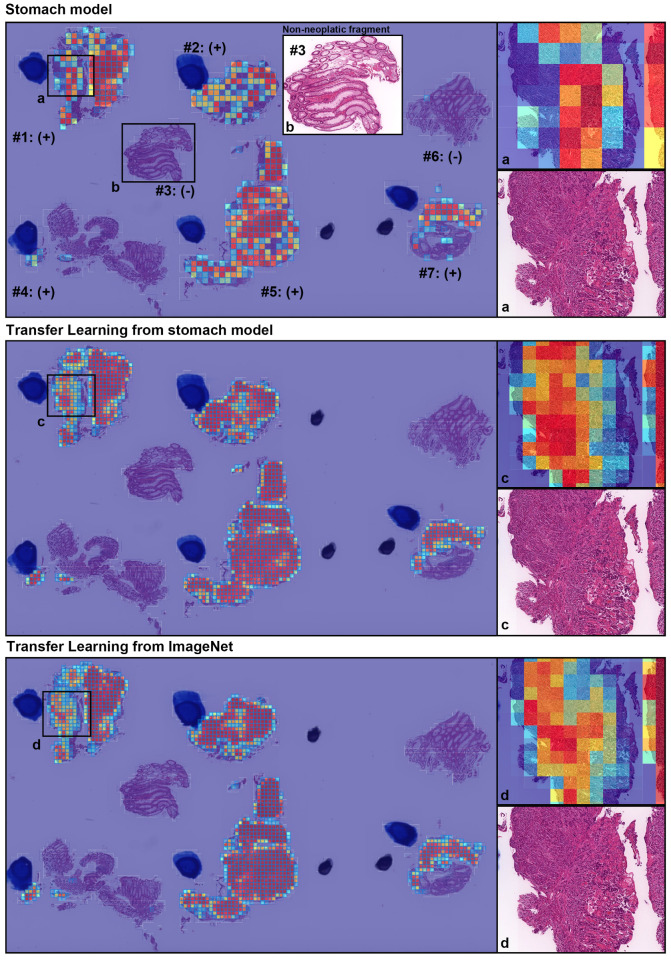
A representative true positive poorly differentiated colorectal ADC case from the endoscopic biopsy test set. Heatmap images show true positive predictions of poorly differentiated ADC cells and they correspond, respectively, to H&E histopathology (a, c, d) using stomach model (**upper panel**), transfer learning from stomach model (**middle panel**), and transfer learning from ImageNet model (**lower panel**). According to the pathological diagnosis provided by surgical pathologists, histopathological evaluation for each tissue fragment is as follows: #1, #2, #4, #5, and #7 were positive for poorly differentiated ADC; #3 and #6 were negative for poorly differentiated ADC. The high magnification image (b) shows representative H&E histology (#3 fragment), which is negative for poorly differentiated ADC.

**Figure 4 diagnostics-11-02074-f004:**
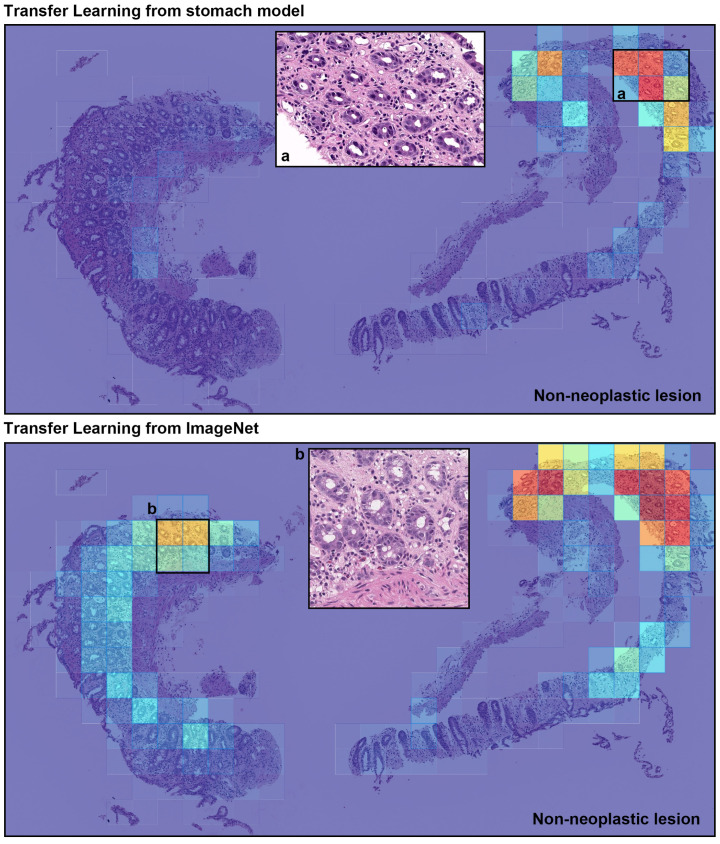
A representative example of poorly differentiated colorectal adenocarcinoma (ADC) false positive prediction output on a case the from endoscopic biopsy test set. Histopathologically, this case is a non-neoplastic lesion (colitis). Heatmap images exhibited false positive prediction of poorly differentiated ADC using transfer learning from stomach model (**upper panel**) and transfer learning from ImageNet model (**lower panel**). The inflammatory tissue with plasma cell infiltration (a and b) is the primary cause of false positive due to its analogous nuclear and cellular morphology to poorly differentiated ADC cells.

**Table 1 diagnostics-11-02074-t001:** Distribution of WSIs in the training, validation, and test sets.

		Poorly Diff. ADC	Well-to-Moderately-Diff. ADC	Adenoma	Non-Neoplastic	Total
Test	Hospital 1	12	125	61	251	449
	Hospital 2	9	41	78	44	172
	Hospital 3	18	20	210	39	287
	Hospital 4	18	74	239	158	489
	Hospital 5	17	144	55	186	402
Training	Hospital 1 & 5	30	60	90	500	680
Validation	Hospital 1 & 5	6	11	20	31	68

**Table 2 diagnostics-11-02074-t002:** ROC and log loss results on the different test sets using the different transfer learning methods.

Method	Source	ROC AUC	Log Loss
Ensemble	combined	0.950 [0.925, 0.971]	0.135 [0.122, 0.148]
TL stomach	combined	0.953 [0.937, 0.966]	0.466 [0.426, 0.506]
TL ImageNet	combined	0.947 [0.923, 0.968]	0.555 [0.511, 0.594]
Stomach model	combined	0.896 [0.862, 0.923]	0.863 [0.814, 0.911]
TL stomach	Hospital 1	0.976 [0.936, 0.997]	0.236 [0.196, 0.271]
TL stomach	Hospital 2	0.964 [0.927, 0.991]	0.459 [0.347, 0.576]
TL stomach	Hospital 3	0.982 [0.966, 0.995]	0.195 [0.143, 0.244]
TL stomach	Hospital 4	0.964 [0.94, 0.983]	0.44 [0.36, 0.515]
TL stomach	Hospital 5	0.932 [0.886, 0.97]	0.949 [0.855, 1.081]
TL ImageNet	Hospital 1	0.903 [0.774, 0.993]	0.325 [0.284, 0.367]
TL ImageNet	Hospital 2	0.973 [0.939, 0.999]	0.613 [0.468, 0.72]
TL ImageNet	Hospital 3	0.983 [0.965, 0.997]	0.268 [0.209, 0.326]
TL ImageNet	Hospital 4	0.97 [0.948, 0.987]	0.48 [0.398, 0.549]
TL ImageNet	Hospital 5	0.923 [0.868, 0.969]	1.085 [0.972, 1.219]
Stomach model	Hospital 1	0.851 [0.739, 0.928]	1.055 [0.953, 1.167]
Stomach model	Hospital 2	0.865 [0.768, 0.951]	0.882 [0.722, 1.032]
Stomach model	Hospital 3	0.924 [0.864, 0.96]	0.607 [0.506, 0.716]
Stomach model	Hospital 4	0.923 [0.843, 0.981]	0.554 [0.475, 0.62]
Stomach model	Hospital 5	0.933 [0.881, 0.972]	1.2 [1.102, 1.326]

**Table 3 diagnostics-11-02074-t003:** Confusion matrix for the ensemble model with a threshold of 0.5.

		Predicted Label	
		Other	Poorly ADC
True label	Other	1572	153
	Poorly ADC	11	63

## Data Availability

Due to specific institutional requirements governing privacy protection, datasets used in this study are not publicly available.

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
