# Peer review of "Deep Learning Models for Poorly Differentiated Colorectal Adenocarcinoma Classification in Whole Slide Images Using Transfer Learning"

_diagnostics, 2021, doi:10.3390/diagnostics11112074_

Round 1
Reviewer 1 Report
The paper submitted by Tsuneki at al is a timely paper for the field and with great interest. The authors trained deep learning models for the classification of diffuse-type 41 colorectal adenocarcinoma in endoscopic biopsy specimen whole slide images (WSIs) showing that that deep learning models could potentially be used for the classification of poorly differentiated ADC. The paper may be published after the authors will insert a section regarding the potential of the method as a routine diagnostic method in CRC diagnostics and how this method can be implemented in hospitals on a large scale-the real clinical benefit.
Author Response
Reviewer 1
The paper submitted by Tsuneki at al is a timely paper for the field and with great interest. The authors trained deep learning models for the classification of diffuse-type 41 colorectal adenocarcinoma in endoscopic biopsy specimen whole slide images (WSIs) showing that that deep learning models could potentially be used for the classification of poorly differentiated ADC. The paper may be published after the authors will insert a section regarding the potential of the method as a routine diagnostic method in CRC diagnostics and how this method can be implemented in hospitals on a large scale-the real clinical benefit.
Response: We have extended the last paragraph in the discussion.
“Poorly differentiated ADC tends to grow and spread more quickly than well and moderately differentiated ADC, and this makes early screening critical for improving patient prognosis. The promising results of this study add to the growing evidence that deep learning models could be used as a tool to aid pathologists in their routine diagnostic workflows, potentially acting as a second screener. One advantage of using an automated tool is that it can systematically handle large amounts of WSIs without potential bias due to fatigue commonly experienced by surgical pathologists. It could also drastically alleviate the heavy clinical burden of daily pathology diagnosis. AI is considered a valuable tool that could transform the future of healthcare and precision oncology.”
Reviewer 2 Report
What was the average number of pixels (or dimensions) in WSIs?
The training/test split was not random, as can be appreciated in Table 1. The test set included 948 WSIs from hospitals not available for training and validation. This enhances the precision of the final estimate of real-life performance.
Please comment on the use of 10-fold cross-validation or LOOCV in your data.
Do you think that using training/inference loop could enhance overfitting?
High AUC values are achieved. However, in the case of cancer diagnostics a false negative may have disastrous consequences. Please provide some information on sensitivity and specificity for the models (not only AUC), which would tell the reader what percentage of neoplasm cases would have been missed by the models (or only by the best model). Please consider high-sensitivity ratio (necessary in practice) and what would be false positive rate then. This is key since the paper comes from a company that commercializes AI solutions for pathology diagnoses. In my view it is not enough that the reader can see this in ROC curves, this is an important aspect to discuss for readers' benefit.
Two pathologists assessed stained biopsy specimens. They were also supervised by a senior pathologist. Pathologists also make errors, just like algorithms. It would be very interesting to see what was the percentage of cases where pathologists did not reach full agreement. These are cases which could prove difficult not only for AI but also problematic for pathologists (as still typically done today).
It was not entirely clear for me from the description (although it is quite detailed and well done) - the training set was balanced by oversampling?
Please correct, "receiver operating characteristic curve" is the typical term (not "reciever operator").
"simply transfer learning method" or "simple transfer learning method"?
The command of English seems excellent.
Another remark, not for the article, but for your future work:
What is presented are models that classify/detect neoplasms. However, are you also able to produce another, completely separate model, trained to detect samples that will prove difficult to classify by the main AI? This way the result from the system would have two components: (1) what is the classification, (2) what is the risk that there is an error, that the AI will not perform well. This would be a "safety valve" for the system, calling in a pathologist to help in diagnosis (in select cases only, possibly ~10%), and maximize patient safety. This would greatly enhance the trust in your products also (if it is not in place already!). (Please note - this is not a suggestion to change your main conclusion, which is: AI can help pathologist. I am just thinking about the future: pathologist aids AI if needed.)
Author Response
Reviewer 2
What was the average number of pixels (or dimensions) in WSIs?
Response: The average number of pixels was 30K x 15 K pixels. We have added this info in the methods section.
The training/test split was not random, as can be appreciated in Table 1. The test set included 948 WSIs from hospitals not available for training and validation. This enhances the precision of the final estimate of real-life performance.
Please comment on the use of 10-fold cross-validation or LOOCV in your data.
Response: While 10 fold cross validation or LOOCV is common in traditional machine learning with limited/small datasets, it tends to be less used in deep learning with large datasets as the test sets are already sufficiently large to give a good estimate. While the number of WSI could appear potentially small, the WSI is composed of thousands of tiles. Using the average number of pixels in each WSI, 30Kx15K px, with stride size of 256x256 px and tile size of 512x512px, we end up with about 30,000x15,000/(256x256)= 6,866 tiles per WSI. The combined total of WSI was 2,547, so we end up with about 17 million tiles. And that’s a large number. It takes a large number of hours to train and test the model, so implementing something like 10 fold cross validation or LOOCV especially is not feasible.
Do you think that using training/inference loop could enhance overfitting?
Response: Overfitting was measured by the performance of the model on the validation set. Based on the results, it did not seem to overfit on the training set.
High AUC values are achieved. However, in the case of cancer diagnostics a false negative may have disastrous consequences. Please provide some information on sensitivity and specificity for the models (not only AUC), which would tell the reader what percentage of neoplasm cases would have been missed by the models (or only by the best model). Please consider high-sensitivity ratio (necessary in practice) and what would be false positive rate then. This is key since the paper comes from a company that commercializes AI solutions for pathology diagnoses. In my view it is not enough that the reader can see this in ROC curves, this is an important aspect to discuss for readers' benefit.
Response: We have added the following in the results section:
“When operating with a high sensitivity threshold (sensitivity 100%), the specificity for the best model on the combined test set was 75%.”
Two pathologists assessed stained biopsy specimens. They were also supervised by a senior pathologist. Pathologists also make errors, just like algorithms. It would be very interesting to see what was the percentage of cases where pathologists did not reach full agreement. These are cases which could prove difficult not only for AI but also problematic for pathologists (as still typically done today).
Response: About 20% of cases were excluded due to disagreement between pathologists during the WSI collection stage. We have updated the methods section with this information.
It was not entirely clear for me from the description (although it is quite detailed and well done) - the training set was balanced by oversampling?
Response: Yes, it was balanced by oversampling. We do mention it in the deep learning models section
Please correct, "receiver operating characteristic curve" is the typical term (not "reciever operator").
Response: We have corrected it, thanks.
"simply transfer learning method" or "simple transfer learning method"?
Response: Simple. We have corrected it, thanks.
The command of English seems excellent.
Response: Thank you.
Another remark, not for the article, but for your future work:
What is presented are models that classify/detect neoplasms. However, are you also able to produce another, completely separate model, trained to detect samples that will prove difficult to classify by the main AI? This way the result from the system would have two components: (1) what is the classification, (2) what is the risk that there is an error, that the AI will not perform well. This would be a "safety valve" for the system, calling in a pathologist to help in diagnosis (in select cases only, possibly ~10%), and maximize patient safety. This would greatly enhance the trust in your products also (if it is not in place already!). (Please note - this is not a suggestion to change your main conclusion, which is: AI can help pathologist. I am just thinking about the future: pathologist aids AI if needed.)
Response: That’s definitely interesting and we are considering doing something like this in the future.